# LOT: Layer-wise Orthogonal Training on Improving $\ell_2$ Certified Robustness

**Xiaojun Xu**   **Linyi Li**   **Bo Li**
University of Illinois Urbana-Champaign
{xiaojun3, linyi2, lbo}@illinois.edu

## Abstract

Recent studies show that training deep neural networks (DNNs) with Lipschitz constraints are able to enhance adversarial robustness and other model properties such as stability. In this paper, we propose a layer-wise orthogonal training method (LOT) to effectively train 1-Lipschitz convolution layers via parametrizing an orthogonal matrix with an unconstrained matrix. We then efficiently compute the inverse square root of a convolution kernel by transforming the input domain to the Fourier frequency domain. On the other hand, as existing works show that semi-supervised training helps improve *empirical* robustness, we aim to bridge the gap and prove that semi-supervised learning also improves the *certified* robustness of Lipschitz-bounded models. We conduct comprehensive evaluations for LOT under different settings. We show that LOT significantly outperforms baselines regarding deterministic $\ell_2$ certified robustness, and scales to deeper neural networks. Under the supervised scenario, we improve the state-of-the-art certified robustness for all architectures (e.g. from 59.04% to 63.50% on CIFAR-10 and from 32.57% to 34.59% on CIFAR-100 at radius $\rho = 36/255$ for 40-layer networks). With semi-supervised learning over unlabelled data, we are able to improve state-of-the-art certified robustness on CIFAR-10 at $\rho = 108/255$ from 36.04% to 42.39%. In addition, LOT consistently outperforms baselines on different model architectures with only 1/3 evaluation time.

## 1 Introduction

Given the wide applications of deep neural networks (DNNs), ensuring their robustness against potential adversarial attacks [7, 21, 31, 32] is of great importance. There has been a line of research providing defense approaches to improve the *empirical* robustness of DNNs [18, 34, 30, 29], and certification methods to *certify* DNN robustness [14, 35, 15, 13, 33]. Existing certification techniques can be categorized as deterministic and probabilistic certifications [14], and in this work we will focus on improving the deterministic $\ell_2$ certified robustness by training 1-Lipschitz DNNs.

Although different approaches have been proposed to empirically enforce the Lipschitz constant of the trained model [26], it is still challenging to strictly ensure the 1-Lipschitz, which can lead to tight robustness certification. One recent work SOC [23] proposes to parametrize the orthogonal weight matrices with the exponential of skew-symmetric matrices (i.e. $W = \exp(V - V^{\mathsf{T}})$). However, such parametrization will be biased when the matrix norm is constrained and the expressiveness is limited, especially when they rescale $V$ to be small to help with convergence. In this work, we propose a layer-wise orthogonal training approach (LOT) by parameterizing the orthogonal weight matrix with an unconstrained matrix $W = (VV^{\mathsf{T}})^{-\frac{1}{2}}V$. In order to calculate the inverse square root for convolution kernels, we will perform Fourier Transformation and calculate the inverse square root of matrices on the frequency domain using Newton's iteration. In our parametrization, the output is agnostic to the input norm (i.e. scaling $V = \alpha V$ does not change the value of $W$). We show that such

36th Conference on Neural Information Processing Systems (NeurIPS 2022).

parametrization achieves higher model expressiveness and robustness (Section 6.2), and provides more meaningful representation vectors (Section 6.3).

In addition, several works have shown that semi-supervised learning will help improve the empirical robustness of models under adversarial training and other settings [4]. In this work, we take the first attempt to bridge semi-supervised training with *certified* robustness based on our 1-Lipschitz DNNs. Theoretically, we show that semi-supervised learning can help improve the error bound of Lipschitz-bounded models. We also lower bound the certified radius as a function of the model performance and Lipschitz property. Empirically, we indeed observe that including un-labelled data will help with the certified robustness of 1-Lipschitz models, especially at a larger radius (e.g. from 36.04% to 42.39% at $\rho = 108/255$ on CIFAR-10).

We conduct comprehensive experiments to evaluate our approach, and we show that LOT significantly outperforms the state-of-the-art in terms of the deterministic $\ell_2$ certified robustness. We also conduct different ablation studies to show that (1) LOT can produce more meaningful features for visualization; (2) residual connections help to smoothify the training of LOT model.

**Technical contributions**. In this work, we aim to train a certifiably robust 1-Lipschitz model and also analyze the certified radius of the Lipschitz bounded model under semi-supervised learning.

- We propose a layer-wise orthogonal training method LOT for convolution layers to train 1-Lipschitz models based on Newton's iteration, and thus compute the deterministic certified robustness for the model. We prove the convergence of Newton's iteration used in our algorithm.

- We derive the certified robustness of lipschitz constrained model under semi-supervised setting, and formally show how semi-supervised learning affects the certified radius.

- We evaluate our LOT method under different settings (i.e. supervised and semi-supervised) on different models and datasets. With supervised learning, we show that it significantly outperforms state-of-the-art baselines, and on some deep architectures the performance gain is over 4%. With semi-supervised learning, we further improve certified robust accuracy by over 6% at a large radius.

## 2  Related Work

**Certified Robustness for Lipschitz Constrained Models**   Several studies have been conducted to explore the Lipschitz-constrained models for certified robustness. [26] first certifies model robustness based on its Lipschitz constant and propose training algorithms to regularize the model Lipschitz. Multiple works [5, 19, 20, 8] have been proposed to achieve 1-Lipschitz during training for linear networks by regularizing or normalizing the spectral norm of the weight matrix. However, when applying these approaches on convolution layers, the spectral norm is bounded by unrolling the convolution into linear operations, which leads to a loose Lipschitz bound [27]. Recently, [2] shows that the 1-Lipschitz requirement is not enough for a good robust model; rather, the gradient-norm-preserving property is important. Besides these training-time techniques, different approaches have been proposed to calculate a tight Lipschitz bound during evaluation. [6] upper bounds the Lipschitz with semi-definite programming while [11] upper bounds the Lipschitz with polynomial optimization. In this work we aim to effectively train 1-Lipschitz convolution models.

**Orthogonal Convolution Neural Networks**   [16] first proposes to directly construct orthogonal convolution operations. Such operations are not only 1-Lipschitz, but also gradient norm preserving, which provides a higher model capability and a smoother training process [2]. BCOP [16] trains orthogonal convolution by iteratively generating $2 \times 2$ orthogonal kernels from orthogonal matrices. [25] proposes to parametrize an orthogonal convolution with Cayley transformation $W = (I - V + V^\intercal)(I + V - V^\intercal)^{-1}$ where the convolution inverse is calculated on the Fourier frequency domain. ECO [36] proposes to explicitly normalize all the singular values [22] of convolution operations to be 1. So far, the best-performing orthogonal convolution approach is SOC [23], where they parametrize the orthogonal convolution with $W = \exp(V - V^\intercal)$ where the exponential and transpose are defined with the convolution operation. However, one major weakness of SOC is that it will rescale $V$ to be small so that the approximation of $\exp$ can converge soon, which will impose a bias on the resulting output space. For example, when $V$ is very small $W$ will be close to $I$. Such norm-dependent property is not desired, and thus we propose a parametrization that is invariant to rescaling. Finally, [24] proposes several techniques for orthogonal CNNs, including a generalized Householder (HH) activation function, a certificate regularizer (CReg) loss, and a last layer normalization (LLN). They can be integrated with our training approach to improve model robustness.

# 3 Problem Setup

## 3.1 Lipschitz Constant of Neural Networks and Certified Robustness

Let $f : \mathbb{R}^d \rightarrow \mathbb{R}^C$ denote a neural network for classification, where $d$ is the input dimension and $C$ is the number of output classes. The model prediction is given by $\arg\max_c f(x)_c$, where $f(x)_c$ represents the prediction probability for class $c$. The Lipschitz constant of the model under $p$-norm is defined as: $\mathrm{Lip}_p(f) = \sup \frac{||f(x_1)-f(x_2)||_p}{||x_1-x_2||_p}$ $\quad \forall x_1, x_2 \in \mathbb{R}^d$. Unless specified, we will focus on $\ell_2$-norm and use $\mathrm{Lip}(f)$ to denote $\mathrm{Lip}_2(f)$ in this work. We can observe that the definition of model Lipschitz aligns with its robustness property - both require the model not to change much w.r.t. input changes. Formally speaking, define $\mathcal{M}_f(x) = \max_i f(x)_i - \max_{j \neq \arg\max_i f(x)_i} f(x)_j$ to be the prediction gap of $f$ on the input $x$, then we can guarantee that $f(x)$ will not change its prediction within $|x' - x| < r$, where

$$r = \mathcal{M}_f(x)/(\sqrt{2}\mathrm{Lip}(f)).$$

Therefore, people have proposed to utilize the model Lipschitz to provide certified robustness and investigated training algorithms to train a model with a small Lipschitz constant.

Note that the Lipschitz constant of a composed function $f = f_1 \circ f_2$ satisfies $\mathrm{Lip}(f) \leq \mathrm{Lip}(f_1) \times \mathrm{Lip}(f_2)$. Since a neural network is usually composed of several layers, we can investigate the Lipschitz of each layer to calculate the final upper bound. If we can restrict each layer to be 1-Lipschitz, then the overall model will be 1-Lipschitz with an arbitrary number of layers.

## 3.2 Orthogonal Linear and Convolution Operations

Consider a linear operation with equal input and output dimensions $y = Wx$, where $x, y \in \mathbb{R}^n$ and $W \in \mathbb{R}^{n \times n}$. We say $W$ is an orthogonal matrix if $WW^\intercal = W^\intercal W = I$ and call $y = Wx$ an orthogonal linear operation. The orthogonal operation is not only 1-Lipschitz, but also norm-preserving, i.e., $||Wx||_2 = ||x||_2$ for all $x \in \mathbb{R}^n$. If the input and output dimensions do not match, i.e., $W \in \mathbb{R}^{m \times n}$ where $n \neq m$, we say $W$ is semi-orthogonal if either $WW^\intercal = I$ or $W^\intercal W = I$. The semi-orthogonal operation is 1-Lipschitz and non-expansive, i.e., $||Wx||_2 \leq ||x||_2$.

The orthogonal convolution operation is defined in a similar way. Let $y = W \circ X$ be an orthogonal operation, where $x, y \in \mathbb{R}^{c \times w \times w}$ and $W \in \mathbb{R}^{c \times c \times k \times k}$. We say $W$ is an orthogonal convolution kernel if $W \circ W^\intercal = W^\intercal \circ W = I_{conv}$ where the transpose here refers to the convolution transpose and $I_{conv}$ denotes the identity convolution kernel. Such orthogonal convolution is 1-Lipschitz and norm-preserving. When the input and output channel numbers are different, the semi-orthogonal convolution kernel $W$ satisfies either $W \circ W^\intercal = I_{conv}$ or $W^\intercal \circ W = I_{conv}$ and it is 1-Lipschitz and non-expansive.

# 4 LOT: Layer-wise Orthogonal Training

In this section, we propose our LOT framework to achieve certified robustness.[1] We will first introduce how LOT layer works to achieve 1-Lipschitz. The key idea of our method is to parametrize an orthogonal convolution $W$ with an un-constrained convolution kernel $W = (VV^\intercal)^{-\frac{1}{2}}V$. Next, we will propose several techniques to improve the training and evaluation processes of our model. Finally, we discuss how semi-supervised learning can help with the certified robustness of our model.

## 4.1 1-Lipschitz Neural Networks via LOT

Our key observation is that we can parametrize an orthogonal matrix $W \in \mathbb{R}^{n \times n}$ with an unconstrained matrix $V \in \mathbb{R}^{n \times n}$ by $W = (VV^\intercal)^{-\frac{1}{2}}V$ [9]. In addition, this equation also holds true in the case of convolution kernel - given any convolution kernel $V \in \mathbb{R}^{c \times c \times k \times k}$, where $c$ denotes the channel number and $k$ denotes the kernel size, we can get an orthogonal convolution kernel by $W = (V \circ V^\intercal)^{-\frac{1}{2}} \circ V$, where transpose and inverse square root are with respect to the convolution operations. The orthogonality of $W$ can be verified by $W \circ W^\intercal = W \circ W^\intercal = I_{conv}$, where $I$ is

---

[1]The code is available at `https://github.com/AI-secure/Layerwise-Orthogonal-Training`.

the identity convolution kernel. This way, we can parametrize an orthogonal convolution layer by training over the un-constrained parameter $V$.

Formally speaking, we will parametrize an orthogonal convolution layer by $Y = (V \circ V^\mathsf{T})^{-\frac{1}{2}} \circ V \circ X$, where $X, Y \in \mathbb{R}^{c \times w \times w}$ and $w \geq k$ is the input shape. The key obstacle here is how to calculate the inverse square root of a convolution kernel. Inspired by [25], we can leverage the convolution theorem which states that the convolution in the input domain equals the element-wise multiplication in the Fourier frequency domain. In the case of multi-channel convolution, the convolution corresponds to the matrix multiplication on each pixel location. That is, let $\mathrm{FFT} : \mathbb{R}^{w \times w} \to \mathbb{C}^{w \times w}$ be the 2D Discrete Fourier Transformation and $\mathrm{FFT}^{-1}$ be the inverse Fourier Transformation. We will zero-pad the input to $w \times w$ if the original shape is smaller than $w$. Let $\tilde{X}_i = \mathrm{FFT}(X_i)$ and $\tilde{V}_{j,i} = \mathrm{FFT}(V_{j,i})$ denote the input and kernel on frequency domain, then we have:

$$\mathrm{FFT}(Y)_{:,a,b} = (\tilde{V}_{:,:,a,b}\tilde{V}^*_{:,:,a,b})^{-\frac{1}{2}}\tilde{V}_{:,:,a,b}\tilde{X}_{:,a,b}$$

in which multiplication, transpose and inverse square root operations are matrix-wise. Therefore, we can first calculate $\mathrm{FFT}(Y)$ on the frequency domain and perform the inverse Fourier transformation to get the final output.

We will use Newton's iteration to calculate the inverse square root of the positive semi-definite matrix $A = \tilde{V}_{:,:,a,b}\tilde{V}^*_{:,:,a,b}$ in a differentiable way [17]. If we initialize $Y_0 = A$ and $Z_0 = I$ and perform the following update rule:

$$Y_{k+1} = \frac{1}{2}Y_k(3I - Z_kY_k), \quad Z_{k+1} = \frac{1}{2}(3I - Z_kY_k)Z_k, \tag{1}$$

$Z_k$ will converge to $A^{-\frac{1}{2}}$ when $||I - A||_2 < 1$. The condition can be satisfied by rescaling the parameter $V = \alpha V$, noticing that the scaling factor will not change the resulting $W$. In practice, we execute a finite number of Newton's iteration steps and we provide a rigorous error bound for this finite iteration scheme in Appendix D to show that the error will decay exponentially. In addition, we show that although the operation is over complex number domain after the FFT, the resulting parameters will still be real domain, as shown below.

**Theorem 4.1.** *Say $J \in \mathbb{C}^{m \times m}$ is unitary so that $J^*J = I$, and $V = J\tilde{V}J^*$ for $V \in \mathbb{R}^{m \times m}$ and $\tilde{V} \in \mathbb{C}^{m \times m}$. Let $F(V) = (VV^*)^{-\frac{1}{2}}V$ be our transformation. Then $F(V) = JF(\tilde{V})J^*$.*

*Proof.* First, notice that $V^T = V^* = J\tilde{V}^*J^*$. Second, we have:

$$(\tilde{V}\tilde{V}^*)^{-1} = J^*(J\tilde{V}\tilde{V}^*J^*)^{-1}J$$
$$= (J^*(J\tilde{V}\tilde{V}^*J^*)^{-\frac{1}{2}}J)^2,$$

so that $(\tilde{V}\tilde{V}^*)^{-\frac{1}{2}} = J^*(J\tilde{V}\tilde{V}^*J^*)^{-\frac{1}{2}}J$. Thus, we have

$$J^*F(V)J = J^*(VV^T)^{-\frac{1}{2}}VJ$$
$$\overset{\text{by } V=J\tilde{V}J^*}{=} J^*(J\tilde{V}J^*J\tilde{V}^*J^*)^{-\frac{1}{2}}J\tilde{V}J^*J = J^*(J\tilde{V}\tilde{V}^*J^*)^{-\frac{1}{2}}J\tilde{V}$$
$$= (\tilde{V}\tilde{V}^*)^{-\frac{1}{2}}\tilde{V}.$$

$\square$

*Remark.* From this theorem it is clear that our returned value $JF(\tilde{V})J^*$ equals to the transformed version of the original matrix $F(V) \in \mathbb{R}^{m \times m}$, and thus is guaranteed to be in the real domain.

**Circular Padding vs. Zero Padding** When we apply the convolution theorem to calculate the convolution $Y = W \circ X$ with Fourier Transformation, the result implicitly uses circular padding. However, in neural networks, zero padding is usually a better choice. Therefore, we will first perform zero padding on both sides of input $X^{pad} = \mathrm{zero\_pad}(X)$ and calculate the resulting $Y^{pad} = W \circ X^{pad}$ with Fourier Transformation. Thus, the implicit circular padding in this process will actually pad the zeros which we padded beforehand. Finally, we truncate the padded part and get our desired output $Y = \mathrm{truncate}(Y^{pad})$. We empirically observe that this technique helps improve the model robustness as shown in Appendix E.3.

**When Input and Output Dimensions Differ**   In previous discussion, we assume that the convolution kernel $V$ has equal input and output channels. In the case $W \in \mathbb{R}^{c_{out} \times c_{in} \times k \times k}$ where $c_{out} \neq c_{in}$, we aim to get a semi-orthogonal convolution kernel $W$ (i.e. $W \circ W^\intercal = I_{conv}$ if $c_{out} < c_{in}$ or $W^\intercal \circ W = I_{conv}$ if $c_{out} > c_{in}$). As pointed out in [9], calculating $W$ with Newton's iteration will naturally lead to a semi-orthogonal convolution kernel when the input and output dimensions differ.

**Emulating a Larger Stride**   To emulate the case when the convolution stride is 2, we follow previous works [23] and use the invertible downsampling layer [10], in which the input dimension $c_{in}$ will be increased by $\times 4$ times. Strides larger than 2 can be emulated with similar techniques if needed.

**Overall Algorithm**   Taking the techniques we discussed before, we can get our final LOT convolution layer. The detailed algorithm is shown in Appendix B. First, we will pad the input to prevent the implicit circular padding mechanism and pad the kernel so that they are in the same shape. Next, we perform the Fourier transformation and calculate the output on the frequency domain with Newton's iteration. Finally, we perform inverse Fourier transformation and return the desired output.

**Comparison with SOC**   Several works have been proposed on orthogonal convolution layers with re-parametrization[25, 23], among which the SOC approach via $W = \exp(V - V^\intercal)$ has achieved the best performance. Compared with SOC, LOT has the following advantages. First, the parametrization in LOT is norm-independent. Rescaling $V$ to $\alpha V$ will not change the resulting $W$. By comparison, $V$ with a smaller norm in SOC will lead to a $W$ closer to the identity transformation. Considering that the norm of $V$ will be regularized during training (e.g. SOC will re-scale $V$ to have a small norm; people usually initialize weight to be small and impose l2-regularization during training), the orthogonal weight space in SOC may be biased. Second, we can see that LOT is able to model any orthogonal kernel $W$ by noticing that $(WW^\intercal)^{-\frac{1}{2}} W$ is $W$ itself; by comparison, SOC cannot parametrize all the orthogonal operations. For example, in the case of orthogonal matrices, the exponential of a skew-symmetric matrix only models the special orthogonal group (i.e. the matrices with +1 determinant). Third, we directly handle the case when $c_{in} \neq c_{out}$, while SOC needs to do extra padding so that the channel numbers match. Finally, LOT is more efficient during evaluation, when we only need to perform the Fourier and inverse Fourier transformation as extra overhead, while SOC needs multiple convolution operations to calculate the exponential. We will further show quantitative comparisons with SOC in Section 6.2.

The major limitation of LOT is its large overhead during training, since we need to calculate Newton's iteration in each training step which takes more time and memory. In addition, we sometimes observe that Newton's iteration is not stable when we perform many steps with 32-bit precision. To overcome this problem, we will pre-calculate Newton's iteration with 64-bit precision during evaluation, as we will introduce in Section 4.2.

### 4.2   Training and Evaluation of LOT

**Smoothing the Training Stage**   In practice, we observe that the LOT layers are highly non-smooth with respect to the parameter $V$ (see Section 6.3). Therefore, the model is difficult to converge during the training process especially when the model is deep. To smooth the training, we propose two techniques. First, we will initialize all except bottleneck layers where $c_{in} = c_{out}$ with identity parameter $V = I$. The bottleneck layers where $c_{in} = c_{out}$ will still be randomly initialized. Second, as pointed out in [12], residual connection helps with model smoothness. Therefore, for the intermediate layers, we will add the 1-Lipschitz residual connection $y = \lambda x + (1 - \lambda) f(x)$. Some work suggests that $\lambda$ here can be trainable [25], while we observe that setting $\lambda = 0.5$ is enough.

**Speeding up the Evaluation Stage**   Notice that after the model is trained, the orthogonal kernel $\tilde{W}$ will no longer change. Therefore, we can pre-calculate Newton's iteration and use the result for each evaluation step. Thus, the only runtime overhead compared with a standard convolution layer during evaluation is the Fourier transformation part. In addition, we observe that using 64-bit precision instead of the commonly used 32-bit precision in Newton's iteration will help with the numerical stability. Therefore, when we pre-calculate $\tilde{W}$, we will first transform $\tilde{V}$ into float64. After Newton's iterations, we transform the resulting $\tilde{W}$ back to float32 type for efficiency.

### 4.3 Semi-supervised Learning

Existing works have shown that semi-supervised learning helps improve empirical robustness [4], while the impact of semi-supervised learning on certified robustness has not been explored yet. We aim to bridge this gap and investigate whether applying semi-supervised learning also benefits the certified robustness of Lipschitz constrained models. As we will discuss in Section 5.1, we theoretically show that the certified robustness of a Lipschitz-bounded model can be improved by semi-supervised learning. In practice, suppose we have a (small) labelled dataset $\mathcal{D}_s = \{(x_i, y_i)\}$ and a (large) unlabelled dataset $\mathcal{D}_u = \{x'_i\}$, we will first train a labeller model $G_{pl}$ over the labelled dataset $\mathcal{D}_s$. Then we use $G_{pl}$ to assign pseudo-labels to the unlabelled dataset to get $\mathcal{D}'_u = \{(x'_i, G_{pl}(x'_i))\}$. Finally, we use the overall dataset $\mathcal{D}_s + \mathcal{D}'_u$ to train the student model $G$ as the final model. We assume that both $G_{pl}$ and $G$ will use the LOT model architecture.

## 5 Theoretical Analysis

In this section, we will introduce our theoretical analysis to show that the certified robustness of Lipschitz-constrained models will benefit from semi-supervised learning. We make similar assumptions with [28] which provides error bound of discrete models in semi-supervised learning, while we will analyze a (Lipschitz-bounded) continuous model and its certified accuracy.

We first introduce *data expansion* proposed in [28], which we will adopt in our analysis. Let $\mathcal{T}$ denote some set of transformations and define $\mathcal{B} \triangleq \{x' | \exists T \in \mathcal{T} \text{ such that } ||x' - T(x)|| < r\}$ to be the set of points within distance $r$ from any of the transformation of $x$. The neighbourhood of $x$ is defined by $\mathcal{N} = \{x' | \mathcal{B}(x) \cap \mathcal{B}(x') \neq \emptyset\}$. Let $P$ denote the data distribution and $P_i$ denote the distribution of data in the $i$-th class. The expansion property is defined as:

**Definition 5.1** ($(a, c)$-expansion, as in [28]). We say that the class-conditional distribution $P_i$ satisfies $(a, c)$-expansion if for all $V \subseteq \mathcal{X}$ with $P_i(V) < a$, the following holds:

$$P_i(\mathcal{N}(V)) \geq \min\{cP_i(V), 1\}$$

if $P_i$ satisfies $(a, c)$-expansion for all classes in $P$, then we say $P$ satisfies $(a, c)$-expansion.

Next, we define the classifier and the loss function in the continuous case. For simplicity, we assume a binary classification task here.

**Definition 5.2** (Continuous model $G$ and loss functions). We assume a binary classification task, where $G^*(x), G_{pl}(x) \in \{0, 1\}$ are discrete labels, and the trained model is $G(x) \in [0, 1]$. We define the loss function $L_{\mathsf{m}}(G, G^*) \in [0, 1]$ to be:

$$L_{\mathsf{m}}(G, G^*) = \mathbb{E}_{x \in \mathcal{X}}[G^*(x)(1 - G(x)) + (1 - G^*(x))G(x)]$$

and the error of $G$ is defined as $\mathrm{Err}_{\mathsf{m}}(G) = L_{\mathsf{m}}(G, G^*)$.

*Remark.* Here we provide a general setting here, and if we assume $G(x) \in \{0, 1\}$, $L_{\mathsf{m}}(G, G^*)$ becomes the 0-1 loss in [28].

Then we define the *marginal* consistency set $S^{\mathsf{m}}_B(G; \delta)$ and loss $R^{\mathsf{m}}_B(G; \delta)$ in the continuous case. The consistency set includes cases in which the model prediction will not change a lot within the neighbourhood, and the consistency loss measures the probability that the model is not consistent over the population.

**Definition 5.3** (Marginal consistency set and consistency loss). We define the set $S^{\mathsf{m}}_B(G; \delta)$ in which the ratio of marginal prediction change on each input $x$ is no larger than $\delta$ within its neighbourhood:

$$S^{\mathsf{m}}_B(G; \delta) = \{x | \frac{G(x)}{G(x')} \geq 1 - \delta \text{ and } \frac{1 - G(x)}{1 - G(x')} \geq 1 - \delta \quad \forall x' \in \mathcal{B}(x)\}.$$

and $R^{\mathsf{m}}_B(G; \delta)$ denotes the marginal consistency loss of the probability that $x$ is not in $S^{\mathsf{m}}_B(G; \delta)$:

$$R^{\mathsf{m}}_B(G; \delta) = \mathbb{P}_{x \in P}[x \notin S^{\mathsf{m}}_B(G; \delta)].$$

*Remark.* (1) We may use $S^{\mathsf{m}}_B(G)$ and $R^{\mathsf{m}}_B(G)$ for abbreviation when the choice of $\delta$ does not lead to confusion; (2) when $\delta = 0$, $S^{\mathsf{m}}_B(G)$ and $R^{\mathsf{m}}_B(G)$ requires that the prediction is exactly the same within the neighbourhood, which are reduced to $S_B(G)$ and $R_B(G)$ defined in [28].

Finally, we define the certified robustness radius CertR$(G)$ as the average radius over the population in which the model keeps its correct prediction:

$$\text{CertR}(G) = \mathbb{E}_{x \in \mathcal{X}} \left[ \sup r \text{ s.t.} \forall ||x' - x||_2 < r, G(x') = G(x) = G^*(x) \right]$$

Note that when $G(x) \neq G^*(x)$, the certified radius $r$ will be 0.

## 5.1 Error Bound and Certified Radius of Lipschitz-bounded Model

Let $\mathcal{M}(G_{pl}) = \{x : G_{pl}(x) \neq G^*(x)\}$ denote the set in which the pseudolabel is wrong. We will have similar separation and expansion assumptions on the data distribution as in [28].

**Assumption 5.1** (Separation). *We assume that $P$ is separable with probability $1 - \mu$ by ground-truth classifier $G^*$, i.e., $R_B^m(G^*; 0) \leq \mu$.*

**Assumption 5.2** (Expansion). *Define $\bar{a} \triangleq \max_i \{P_i(\mathcal{M}(G_{pl}))\}$ to be the maximum fraction of incorrectly pseudolabeled examples in any class. We assume that $\bar{a} < 1/3$ and $P$ satisfies $(\bar{a}, \bar{c})$-expansion for $\bar{c} > 3$. We express our bounds in terms of $c \triangleq \min(1/\bar{a}, \bar{c})$.*

With these assumptions and let $\delta \in [0, \frac{1}{c}]$, we can compare the performance of $G$ and $G_{pl}$ (i.e. the performance with and without semi-supervised learning) with the following theorem:

**Theorem 5.1** (Error bound with semi-supervised learning). *Suppose Assumption 5.1 and 5.2 holds and let $\hat{G}$ be a minimizer of the following loss function:*

$$\hat{G} = \arg\min_G L(G) \triangleq \frac{c+3}{c-1} L_m(G, G_{pl}) + \frac{2c+2}{c-1} R_B^m(G; \delta) - Err(G_{pl})$$

*Then we can upper bound the error of $\hat{G}$ by:*

$$Err_m(\hat{G}) \leq \frac{4}{c-1} Err(G_{pl}) + \frac{2c+2}{c-1} \mu$$

In addition, we can bound the certified robustness radius w.r.t. the Lipschitz constant as below:

**Theorem 5.2** (Certified radius with semi-supervised learning). *Suppose Assumption 5.2 holds. Then the certified radius of a model $G$ can be bounded by:*

$$CertR(G) \geq \frac{0.5 - \frac{c+3}{c-1} L_m(G, G_{pl}) - \frac{2c+2}{c-1} R_B^m(G; \delta) + Err(G_{pl})}{Lip(G)}.$$

The proofs of Theorem 5.1 and 5.2 are shown in Appendix C.

**Discussion** From Theorem 5.1, we observe that the benefits of semi-supervised learning depend on the error of $G_{pl}$ and data property. If the data expands and separates well and $G_{pl}$ is accurate, $c$ is large and $\mu$ is small, which means $Err_m(\hat{G})$ will be smaller than $Err(G_{pl})$. Note that $\mu$ is usually small and we assume $c > 3$, so $Err_m(\hat{G}) < Err(G_{pl})$ will hold true in most cases, indicating that semi-supervised learning helps improve the model performance. From Theorem 5.2, we can observe that given the data distribution and $G_{pl}$, the certified radius of the student model CertR$(G)$ can be lower-bounded by 1) how well $G$ learns from $G_{pl}$ ($L_m(G, G_{pl})$); 2) how consistent $G$ is ($R_B^m(G; \delta)$). These two factors also correspond to the losses in the actual training process - the cross-entropy loss measures how well $G$ learns from $G_{pl}$ and the Lipschitz constraint measures local consistency for a Lipschitz-bounded model[2].

# 6 Evaluation

In this section, we will evaluate our method LOT on different datasets and models compared with the state-of-the-art baselines based on the deterministic $\ell_2$ certified robustness. We show that on both CIFAR-10 and CIFAR-100, LOT achieves significantly higher certified accuracy under different radii and settings with more efficient evaluation time. In the meantime, we conduct a series of ablation studies to analyze the representation power of our LOT, the error control of Newton's iterations, and the effects of residual connections. We show that under the semi-supervised setting, the certified accuracy of LOT is further improved and outperforms the baselines.

---

[2]This means that, with smaller Lipschitz, the model is more likely to be consistent within neighbourhood. In addition, the CReg loss will also improve local consistency by maximizing the prediction gap.

Table 1: Certified accuracy of 1-Lipschitz model with CReg loss and HH activation on CIFAR-10/100 in supervised setting. The LLN technique is applied on the CIFAR-100 dataset.

| Model | Conv. Type | CIFAR-10 | | | | CIFAR-100 | | | | Mean Evaluation Time (sec) |
|---|---|---|---|---|---|---|---|---|---|---|
| | | Vanilla Accuracy | Certified Accuracy at $\rho=$ | | | Vanilla Accuracy | Certified Accuracy at $\rho=$ | | | |
| | | | 36/255 | 72/255 | 108/255 | | 36/255 | 72/255 | 108/255 | |
| LipConvnet-5 | SOC | 75.31% | 60.37% | 45.62% | 32.38% | 45.82% | 32.99% | 22.48% | 14.79% | 2.285 |
| | LOT | **76.34%** | **62.07%** | **47.52%** | **33.99%** | **48.38%** | **34.77%** | **23.38%** | **15.44%** | **1.411** |
| LipConvnet-10 | SOC | 76.23% | 62.57% | 47.70% | 34.15% | 47.07% | 34.53% | 23.50% | 15.66% | 3.342 |
| | LOT | **76.50%** | **63.12%** | **48.59%** | **35.65%** | **48.70%** | **34.82%** | **23.86%** | **15.93%** | **1.563** |
| LipConvnet-15 | SOC | 76.39% | 62.96% | 48.47% | 35.47% | 47.61% | 34.54% | 23.16% | 15.09% | 4.105 |
| | LOT | **76.86%** | **63.84%** | **49.18%** | **36.35%** | **48.99%** | **34.90%** | **24.39%** | **16.37%** | **1.627** |
| LipConvnet-20 | SOC | 76.34% | 62.63% | 48.69% | 36.04% | 47.84% | 34.77% | 23.70% | 15.84% | 5.142 |
| | LOT | **77.12%** | **64.30%** | **49.49%** | **36.34%** | **48.81%** | **35.21%** | **24.37%** | **16.23%** | **1.962** |
| LipConvnet-25 | SOC | 75.21% | 61.98% | 47.93% | 34.92% | 46.87% | 34.09% | 23.41% | 15.61% | 6.087 |
| | LOT | **76.83%** | **64.49%** | **49.80%** | **37.32%** | **48.93%** | **35.23%** | **24.33%** | **16.59%** | **2.387** |
| LipConvnet-30 | SOC | 74.23% | 60.64% | 46.51% | 34.08% | 46.92% | 34.17% | 23.21% | 15.84% | 6.927 |
| | LOT | **77.12%** | **64.36%** | **49.98%** | **37.30%** | **49.18%** | **35.54%** | **24.24%** | **16.48%** | **2.755** |
| LipConvnet-35 | SOC | 74.25% | 61.30% | 47.60% | 35.21% | 46.88% | 33.64% | 23.34% | 15.73% | 7.870 |
| | LOT | **76.91%** | **63.55%** | **49.05%** | **36.19%** | **48.25%** | **34.99%** | **24.13%** | **16.25%** | **3.193** |
| LipConvnet-40 | SOC | 72.59% | 59.04% | 44.92% | 32.87% | 45.03% | 32.57% | 22.37% | 14.76% | 8.668 |
| | LOT | **76.75%** | **63.50%** | **49.07%** | **36.06%** | **48.31%** | **34.59%** | **23.70%** | **15.94%** | **3.595** |

## 6.1 Experiment Setup

**Baseline - SOC with CReg loss, HH activation and LLN technique**    In the evaluation, we will mainly compare with SOC [23], as SOC has been shown to outperform other 1-Lipschitz models in terms of certified robustness. SOC parameterizes an orthogonal layer with $W = \exp(V - V^\intercal)$ where $V$ is the convolution kernel and the transpose and exponential are with respect to the convolution operation. Following their setting, we will focus on the CIFAR-10 and CIFAR-100 datasets and the model architecture varies from LipConvnet-5 to LipConvnet-40.

[24] proposes three techniques to improve SOC networks . First, they observe that adding a Certificate Regularization (CReg) loss can help with the certified performance at larger radius: $\ell_{CReg}(x, y) = -\gamma\text{ReLU}\big((f(x)_y - \max_{i \neq y} f(x)_i)/\sqrt{2}\big)$. Second, they propose a Householder (HH) activation layer, which is a generalization of the standard GroupSort activation. Finally, for tasks with large number of classes (e.g. CIFAR-100), they propose Last Layer Normalization(LLN) to only normalize the final output layer instead of orthogonalize. Since these techniques are agnostic to the type of convolution operation, we will further integrate them with LOT for comparison.

**Implementation Details**    We will train and evaluate the LOT network under supervised and unsupervised scenarios. Following previous works [23], we focus on the CIFAR-10 and CIFAR-100 datasets and the model architecture varies from LipConvnet-5 to LipConvnet-40. In semi-supervised learning, we use the 500K data introduced in [4] as the unlabelled dataset. To train the LOT network, we will train the model for 200 epochs using a momentum SGD optimizer with an initial learning rate 0.1 and decay by 0.1 at the 100-th and 150-th epochs. We use Newton's iteration with 10 steps which we observe is enough for convergence (see Appendix E.4). When CReg loss is applied, we use $\gamma = 0.5$; when HH activation is applied, we use the version of order 1. We add the residual connection with a fixed $\lambda = 0.5$ for LOT; for SOC, we use their original version, as we observe that residual connections even hurt their performance (see discussions in Section 6.3). We show the certified accuracy at radius $\rho \in \{\frac{36}{255}, \frac{72}{255}, \frac{108}{255}\}$. For the evaluation time comparison, we show the runtime taken to do a full pass on the testing set evaluated on an NVIDIA RTX A6000 GPU.

## 6.2 Supervised Scenario

We show the results of supervised learning on CIFAR-10 and CIFAR-100 in Table 1. We can observe that LOT significantly outperforms baselines on both vanilla accuracy and certified accuracy for different datasets. In particular, we observe that the improvement is more significant for deeper models. We owe it to the reason that our LOT layer has a better expressiveness and unbiased parametrization. "Mean evaluation time" column in Table 1 records the evaluation time per instance averaged between CIFAR-10 and CIFAR-100 models. Since architectures of CIFAR-10 and CIFAR-100 models differ only in the last linear year, the evaluation time is almost the same. We observe that LOT has a better efficiency during the evaluation stage, since LOT pre-calculates Newton's iteration while SOC calculates the exponential in each iteration. A similar conclusion can be observed from results without CReg loss, HH activation and LLN, as shown in Table 6 and 7 in Appendix E.5.

Table 2: Performance comparison of 1-Lipschitz networks with and without residual connections.

| Model | Conv. Type | Residual Connection | Vanilla Accuracy | Certified Accuracy at $\rho =$ | | |
|---|---|---|---|---|---|---|
| | | | | 36/255 | 72/255 | 108/255 |
| LipConvnet-20 | SOC | × | 76.90% | 61.87% | 45.79% | 31.08% |
| | | ✓ | 76.97% | 61.76% | 45.46% | 30.29% |
| | LOT | × | 77.16% | 62.14% | 45.78% | 30.95% |
| | | ✓ | **77.86%** | **63.54%** | **47.15%** | **32.12%** |

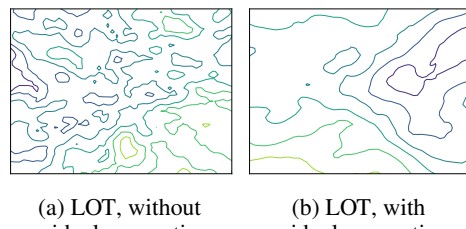

Figure 1: Visualizing the features in the last hidden layer of LipConvnet-20 for SOC (left) and LOT (right). Each image corresponds to one randomly chosen neuron from the last hidden layer and is optimized to maximize the value of the neuron.

## 6.3 Ablation Studies

In this section, we will perform several ablation studies for LOT. Unless specified, we will use the deep model LipConvnet-20 for evaluation. We use the model without CReg loss and HH activation so that we can see the comparison under the standard CNN setting.

**Representation analysis** Recent works suggest that adversarially robust models will have a good regularization-free feature visualization [1]. We show the visualization of 4 neurons in the last hidden layer of LipConvNet-20 for both SOC and LOT in Figure 1. More figures are shown in Appendix E.1 The visualization process is as follows: given a chosen neuron, we will start from a random image and take 500 gradient steps to maximize the neuron value with step size 1.0 and decay factor 0.1. We can see that LOT indeed generates more meaningful features. We attribute it to the good expressiveness power of our model. This indicates that our approach indeed leads to better representations.

**Error Control of Newton's Iterations** To see how our Newton's iteration approximates the inverse square root, we visualize the maximum singular value ($\sigma_{max}$) of LOT layers in Appendix E.4. We observe that all the resulting layers have a strictly $< 1$ Lipschitz. Therefore, we can safely conclude that the overall LOT network is 1-Lipschitz.

**Effect of residual connections** Empirically, we observe that LOT is non-smooth and therefore we need the residual connection to smoothify the model. To verify this phenomenon, we adopt the visualization approach in [12] and show the loss surface in Figure 2. We can see that LOT without residual connection is highly non-smooth, while after adding residual connections the model becomes much smoother. By comparison, we observe that SOC networks with and without residual connection are similar (see Appendix E.2). We hypothesize the reason to be that the exponential parametrization $W = \exp(A)$ enables residual property implicitly, i.e.,

$$\exp(A) \circ X = \underline{X + A \circ X} + \frac{1}{2} A \circ A \circ X + \dots$$

(a) LOT, without residual connection.

(b) LOT, with residual connection.

Figure 2: The loss landscape [12] with respect to the parameters of LOT with and without residual connections. The figure is plotted by calculating the loss contour on two randomly chosen directions of parameters.

where the first two terms are essentially the format of residual connection. We show the results of models with and without residual connections in Table 2. We can observe that our LOT indeed improves with the residual connection, while the performance remains similar or even worse for SOC. Therefore, in the evaluation, we use the LOT with residual connection and SOC without it.

**Robustness against $\ell_\infty$ Empirical Attack** To evaluate the robustness of our model against attacks in other $\ell_p$ norms, we evaluate the model against standard $\ell_\infty$ PGD attack[18] with $\epsilon = 8/255$ and 50 steps. We show the results on CIFAR-10 with CReg loss and HH activation in Table 3. We can observe that we still achieve better empirical robustness on the LipConvNet compared with SOC.

Table 3: Empirical robust accuracy of 1-Lipschitz model with CReg loss and HH activation on CIFAR-10 against $\ell_\infty$-PGD attack at $\epsilon = 8/255$.

| Model | Conv. | Vanilla accuracy | Robust accuracy |
|---|---|---|---|
| LipConvnet-5 | SOC | 75.31% | 27.18% |
| | LOT | **76.34%** | **27.19%** |
| LipConvnet-10 | SOC | 76.23% | 27.95% |
| | LOT | **76.50%** | **28.71%** |
| LipConvnet-15 | SOC | 76.39% | 27.84% |
| | LOT | **76.86%** | **29.10%** |
| LipConvnet-20 | SOC | 76.34% | 26.17% |
| | LOT | **77.12%** | **29.45%** |
| LipConvnet-25 | SOC | 75.21% | 28.93% |
| | LOT | **76.83%** | **29.55%** |
| LipConvnet-30 | SOC | 74.23% | 27.44% |
| | LOT | **77.12%** | **28.92%** |
| LipConvnet-35 | SOC | 74.25% | 14.09% |
| | LOT | **76.91%** | **29.32%** |
| LipConvnet-40 | SOC | 72.59% | 11.68% |
| | LOT | **76.75%** | **28.67%** |

Table 4: Performance of 1-Lipschitz networks on the CIFAR-10 dataset with semi-supervised training. Results of different architectures are shown in Table 8 in Appendix E.6.

| Model | Conv. Type | Setting | Vanilla Accuracy | Certified Accuracy at $\rho =$ | | |
|---|---|---|---|---|---|---|
| | | | | 36/255 | 72/255 | 108/255 |
| LipConvnet-20 | SOC | Supervised | 76.34% | 62.63% | 48.69% | 36.04% |
| | | Semi-supervised | 70.95% | 61.72% | 51.78% | 42.01% |
| | LOT | Supervised | **77.12%** | **64.30%** | 49.49% | 36.34% |
| | | Semi-supervised | 71.86% | 62.86% | **52.24%** | **42.39%** |

## 6.4 Semi-supervised Learning Scenario

We show the results of semi-supervised learning for one architecture with CReg loss and HH activation in Table 4. The full results are shown in Table 8 and 9 in Appendix E.6. We can observe that, with semi-supervised learning, the vanilla accuracy will drop slightly, which will impact the certified accuracy at a small radius ($\rho = 36/255$). However, the certified accuracy at a large radius will be improved, and the gap is more significant at a larger radius (e.g. $\rho = 108/255$). We improve the previous state-of-the-art certified accuracy by over 6% at $\rho = 108/255$. This indeed shows that semi-supervised learning can help improve certified robustness. We empirically observe that semi-supervised learning does not help much on CIFAR-100. We owe it to the reason that the vanilla accuracy of the teacher model on CIFAR-100 is low (less than 50%). Thus, according to Assumption 5.2 and Theorem 5.2, when $c$ is low, the certified accuracy would be low.

## 7 Conclusion

In this work, we propose an orthogonal convolution layer LOT and build a 1-Lipschitz convolution network. We show that LOT network outperforms the previous state-of-the-art in certified robustness. We also show that semi-supervised learning can further help with the robustness both theoretically and experimentally.

## Acknowledgements

This work is partially supported by the NSF grant No.1910100, NSF CNS No.2046726, C3 AI, and the Alfred P. Sloan Foundation.

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
