# OpenReview forum: "LOT: Layer-wise Orthogonal Training on Improving l2 Certified Robustness"
_NeurIPS.cc/2022/Conference — NeurIPS 2022 Accept_

### Official Review · Reviewer_YdF6 · 2022-07-09

**Rating:** 6
**Confidence:** 4
**Soundness:** 3 good
**Presentation:** 3 good
**Contribution:** 2 fair

**Summary:**

The paper proposes a method to train a 1-Lipschitz neural network to achieve improved certified L2 robustness by ensuring that the weight matrix in each layer is orthogonal. The weight matrix W is parameterized using an unconstrained matrix V as W = (V V^T)^(-1/2) V to ensure orthogonality. This parameterization is invariant to rescaling of the unconstrained matrix V, which the authors show leads to better robustness and expressiveness of the model. The paper also shows how to make convolution operations orthogonal. Lipschitz constraints have been shown to improve the empirical robustness of a model. The aim of this work is to show that such models can achieve better certified robustness as well under the semi-supervised learning setting.

**Questions:**

1. The matrix V does not really seem to be unrestricted since it needs to satisfy the condition: || I - VV^T|| < 1. Also, it is not clear to me how scaling V can always make it satisfy the condition as stated in line 154, e.g., consider V = zero matrix in a fully connected layer.
2. How do we initialize V? Do we set it to a particular type of matrix such as an orthogonal matrix?
3. How do we ensure the condition on V is satisfied after each training step?


**Limitations:**

There does not seem to be any negative societal impact of this work.

**Strengths And Weaknesses:**

Originality: The methods used in the paper such as parameterizing orthogonal weight matrices with unconstrained matrices have been considered before. However, the paper presents a new kind of parameterization that has desirable properties such as scale invariance.

Quality: The paper considers several different convolution operations such as circular padding, zero padding, larger strides, etc and shows ways to orthogonalize them. The paper performs a theoretical analysis to demonstrate the benefits of semi-supervised learning for certified robustness. The method has been evaluated and compared with a previous paper in this area for different model architectures.

Clarity: The paper is clearly written and easy to read.

Significance: The method achieves improved performance over the existing approach of Skew Orthogonal Convolutions (SOC).

---

> ### Author Response · Authors · 2022-08-02
> **Author Response**
>
> We thank the reviewer for appreciating our work and providing insightful comments and suggestions. We have provided responses to the questions below and updated our revision following the suggestions.
>
>
>
> > [Q1] The matrix V does not really seem to be unrestricted since it needs to satisfy the condition: || I - VV^T|| < 1. Also, it is not clear to me how scaling V can always make it satisfy the condition as stated in line 154, e.g., consider V = zero matrix in a fully connected layer.
>
> **Response**: By scaling $V$, we can always get a matrix $V’ = \alpha V$ such that all the singular values of $V’$ satisfies $0 \leq \sigma(V’) < 1$. This indicates that $|| I - VV^T || < 1$ and therefore the condition is satisfied. In addition, we notice that such scaling will not affect the result of $W$, as we discussed in L181-L186.
>
> > [Q2] How do we initialize V? Do we set it to a particular type of matrix such as an orthogonal matrix?
>
> **Response**: Thanks for the comment. As introduced in L204-L205, we will initialize the $V$ of bottleneck layers with standard random initialization and other layers to be identity convolution kernel $I$. The initialization of $V$ will not violate the condition $|| I - VV^T|| < 1$, as we will do the scaling operation.
>
> > [Q3] How do we ensure the condition on V is satisfied after each training step?
>
> **Response**: Thanks for the question. As introduced above, we will scale $V$ so that the condition is satisfied and such scaling will not affect the result. We will add related discussions in the revision to make it clear.

---

### Official Review · Reviewer_zFGT · 2022-07-10

**Rating:** 6
**Confidence:** 3
**Soundness:** 3 good
**Presentation:** 4 excellent
**Contribution:** 2 fair

**Summary:**

This work proposed a method that utilizes the convolution theorem and Newton iteration to achieve 1-Lipschitz in convolutional layers of CNN and theoretically justified the effectiveness of semi-supervised training on improving empirical robustness.

**Questions:**

My major concerns of the paper are stated in the weaknesses section. I would appreciate it and am willing to raise my score if the authors could illustrate more the comparisons between at least [1][2][3] and the proposed method, both theoretically (could be intuitive) and empirically.

[1] Asher Trockman and J Zico Kolter. Orthogonalizing convolutional layers with the Cayley transform. In International Conference on Learning Representations, 2021.

[2] Liu, et al. Convolutional Normalization: Improving Deep Convolutional Network Robustness and Training. In Advances in Neural Information Processing Systems, 2021.

[3] Huang, et al. Controllable orthogonalization in training DNNs. In Proceedings of the IEEE/CVF Conference on Computer Vision and Pattern Recognition, 2020.

**Strengths And Weaknesses:**

Strengths: The overall presentation of the work is clear and easy to follow. Although the reviewer did not check all the proofs, the theoretical side of the paper is complete and carefully organized. The paper also talks about the actual implementation (e.g., dealing with padding/stride, etc) and the limitations of the method in great detail.

Weaknesses:
1. The novelty of the method is somewhat limited. For example, calculating things in the Fourier domain is not new, at least the Cayley transform paper [1] cited in the work and [2] utilize similar idea; also, using Newton iteration to approximate matrix inversion is also explored in previous works [3] of the same domain.

2. The experimental results are not super convincing. The paper mainly conducts experiments to compare with SOC and claims that "SOC has been shown to outperform other 1-Lipschitz models". However, the SOC paper mostly only compares with BCOP. I believe a more comprehensive comparison here would make the paper more convincing (at least compare with [1] and [3] which inspired this work?)

---

> ### Author Response · Authors · 2022-08-02
> **Author Response [2/2]**
>
> > [Q2] **Comparison with baselines** (The paper mainly conducts experiments to compare with SOC and claims that "SOC has been shown to outperform other 1-Lipschitz models". However, the SOC paper mostly only compares with BCOP. I believe a more comprehensive comparison here would make the paper more convincing (at least compare with [1] and [3] which inspired this work?)
>
> **Response**: We thank the reviewer for pointing out these related works. We conduct additional experiments on [1] and [3] and show the performance as in the table below following the suggestion. We can observe that LOT outperforms these related works. We provide the discussion of the related works [1,2,3] after the table.
>
> |                      | Vanilla acc | $\rho$=36/255 | $\rho$=72/255 | $\rho$=108/255 |
> | -------------------- | ----------- | ------------- | ------------- | -------------- |
> | LipConvNet-5,ONI     | 69.16%      | 44.74%        | 22.65%        | 9.22%          |
> | LipConvNet-5,Cayley  | 72.37%      | 55.92%        | 38.65%        | 24.27%         |
> | LipConvNet-5,LOT     | **77.20%**  | **61.76%**    | **44.45%**    | **29.61%**     |
> |                      |             |               |               |                |
> | LipConvNet-10,ONI    | 55.15%      | 19.79%        | 3.67%         | 0.40%          |
> | LipConvNet-10,Cayley | 74.30%      | 57.99%        | 40.75%        | 25.93%         |
> | LipConvNet-10,LOT    | **77.30%**  | **62.54%**    | **46.03%**    | **30.64%**     |
> |                      |             |               |               |                |
> | LipConvNet-15,ONI    | 49.05%      | 12.78%        | 1.46%         | 0.06%          |
> | LipConvNet-15,Cayley | 71.92%      | 54.55%        | 37.67%        | 23.50%         |
> | LipConvNet-15,LOT    | **77.34%**  | **63.40%**    | **46.54%**    | **31.75%**     |
> |                      |             |               |               |                |
> | LipConvNet-20,ONI    | 45.04%      | 7.81%         | 0.31%         | 0.01%          |
> | LipConvNet-20,Cayley | 68.87%      | 51.88%        | 35.56%        | 21.72%         |
> | LipConvNet-20,LOT    | **77.86%**  | **63.54%**    | **47.15%**    | **32.12%**     |
>
> - The Cayley transformation work ([1]) has the same goal with us to build 1-Lipschitz CNNs to get certified robustness. In the paper *Improved deterministic l2 robustness on CIFAR-10 and CIFAR-100 (Singla et al., ICLR 2021)*, which is a follow-up work of SOC, the authors compare the performance between Cayley and SOC and show that SOC has the better performance in their Table 7. Therefore, we claim that “SOC has been shown to outperform other 1-Lipschitz models”. We use the results from their paper as a comparison in the table above, denoted as “Cayley”. We can observe that LOT outperforms Cayley with a noticable gap.
> - The Convolution Normalization work ([2]) performs normalization on the convolution kernel in order to improve the *empirical* robustness of models. Although their approach will reduce the Lipschitz constant of the each layer, it is still larger than 1 and thus the overall Lipschitz is very large (see Fig. 2 in their paper). Therefore, their approach cannot achieve a good performance in our task to provide certified robustness of the model.
> - The orthogonal training work ([3]) proposes to orthogonalize a linear layer with Newton’s iteration. However, when the authors apply their approach to conv layers in [3], they simply unroll the convolution kernel into a linear matrix, which will add an extra factor of $k$ to the Lipschitz bound. Therefore, their Lipschitz bound is loose, as we introduce in L67-L70 (actually, we have confirmed with the authors of [3] through email that their bound on conv layers is loose). We show the results that directly using their conv layers in the table above, denoted as “...,ONI”. We can observe that LOT outperforms ONI with a significant gap. Note that ONI cannot perform well in deeper models because its Lipschitz bound is loose.We will add related discussion to our revision, and thank you for the suggestions to help improve our work.

---

> > ### Comment · Reviewer_zFGT · 2022-08-08
> > **Comments on the Response**
> >
> > Thanks for addressing my concerns and showing the additional experiments!

---

> ### Author Response · Authors · 2022-08-02
> **Author Response [1/2]**
>
> We thank the reviewer for the insightful comments and suggestions. We have provided our answers below and updated our paper following the suggestions.
>
> > [Q1] **Novelty** (The novelty of the method is somewhat limited. For example, calculating things in the Fourier domain is not new, at least the Cayley transform paper [1] cited in the work and [2] utilize similar idea; also, using Newton iteration to approximate matrix inversion is also explored in previous works [3] of the same domain.)
>
> **Response**: Thanks for the comment! We agree that both Newton’s iteration and Fourier transformation is not a novel technique in the study of model Lipschitz. Nevertheless, we propose a state-of-the-art Lipschitz network where we utilize the techniques in a proper manner and with carefully designed details (e.g. we are the first to investigate the difference between zero-padding and circular padding when using the Fourier domain in L157-164, which is important to model performance). In addition, we are also the first to study the impact of semi-supervised learning in Lipschitz networks and we provide both theoretical and empirical results.

---

### Official Review · Reviewer_miXu · 2022-07-11

**Rating:** 6
**Confidence:** 3
**Soundness:** 2 fair
**Presentation:** 2 fair
**Contribution:** 2 fair

**Summary:**

This paper proposes a training framework named Layer-wise Orthogonal Training (LOT) that aims to train 1-Lipschitz convolution layers effectively and improves the certified robustness of Lipschitz-bounded models. The paper also proves that semi-supervised learning can benefit the robustness of Lipschitz-bounded models. The key idea of LOT is to utilize unconstrained matrices to approximate an orthogonal matrix and to speed up the inverse computation by mapping the matrices to the FFT domain. The experiments are done on CIFAR-10/100, taking LipConvnet with a different number of layers as the backbone, showing LOT outperforms its directly competing framework called SOC. The main contributions are:
1.	LOT parametrizes the orthogonal weight matrix W with an unconstrained matrix V, and uses FFT to calculate the inverse square root of matrices faster in the frequency domain.
2.	This paper derives the certified robustness of the Lipschitz constrained model under the semi-supervised setting.
3.	LOT improves certified L2 robust accuracy against SOC.


**Questions:**

Further comments:
1. Sect. 5 on the theory is quite involved in terms of math, while its conclusion/discussion is rather trivial. It sounds a bit like “if it’s good then it’s good”.
2. The same issue happens for Fig. 1 and its discussion in Sect. 6.3, too. Why are the features from LOT “more meaningful” than those from SOC? Simply because the patterns have higher amplitudes and fluctuations?
3. Upper half of p.4 shows the FFT-domain operations. I suppose the matrices are holding complex numbers? How does that affect the complexity and stability of the Newton iterations? And how do you guarantee real matrices when mapped back to the spatial domain?
4. I wonder if your Lipschitz constraint can be imposed on the doubly block Toeplitz convolution operator in the OCNN paper [24] and alike? So that your convolution doesn’t need to be in multiple steps with high complexity and low numerical stability?
5. One very key issue I have spotted from the paper is the lurking instability in training/inference, as the authors need to deliberately jet up the accuracy to FP64 during training/inference and then trim back to FP32. This is likely due to the inverse operation which can become ill-conditioned. This may also be why only CIFAR10/100 examples are used, without moving on to the larger sizes like 224^2 ImageNet examples? In fact, do you have larger examples beyond 32^2 input sizes?
6. Following from above, it sounds impossible to quantize your network to INT8, INT4 or even binary as in other CNN compression works?
7. Last but actually the most important, the experiments are only conducted on LipConvnet which is NOT conventionally used in practice. People in the field know very well that robustness is very much architecture-dependent, where transformers exhibit both high accuracy and robustness. The authors may want to study the leaderboard at http://robust.art/ , to see how real-world networks perform under field tests on ImageNet-class inputs. This in turn renders the LipConvnet characterization of this work purely theoretical yet, frankly, with little practical value as the generalization of the results to the more common modern architectures is doubtful (ConvNeXt, Transformers, etc.).

Editorial:
1. It is inappropriate to write V=\alpha V in several places throughout the paper.
2. line 65. “[23] first certify …”, certifies
3. line 70, “Recently, [2] show that …”, shows


**Limitations:**

See my questions above.

**Strengths And Weaknesses:**

Strengths:
1. The scale-invariant convolution kernel parametrization can improve robustness further than SOC.
2. The improvement in the Lipschitz is backed by analytical derivations.

Weaknesses:
1. The derivation of this paper is specific to l2 attacks only, while attacks are conventionally taking l1, l2 and l_\infty norms. In fact, the benchmarked SOC paper also shows empirical robustness in other norms.
2. Only CIFAR10/100 test results are provided, which limits the practicality of the framework.
3. In Sect. 3.2, the semi-orthogonality in the linear operation context is fine, but then its convolution counterpart sounds hand-waving. Is the latter mathematically equivalent to its linearized counterpart? Can you use a toy example to show how to bridge the former to the latter? If they are not directly corresponding, then a proof that conv(W, W^T)=I_{conv} is needed.
4. The definition of W in terms of V requires multiple steps of convolution plus inversion in the FFT domain (and thus complex numbers). I wonder, once V is trained and fixed, why can’t W be computed exactly in terms of V, and then the convolution be done in one step using W? Is that due to non-exactness in the orthogonality or what?

---

> ### Author Response · Authors · 2022-08-02
> **Author Response [3/3]**
>
> > [Q9] **Comparison with OCNN** (I wonder if your Lipschitz constraint can be imposed on the doubly block Toeplitz convolution operator in the OCNN paper [24] and alike? So that your convolution doesn’t need to be in multiple steps with high complexity and low numerical stability?)
>
> **Response**: Thanks for the insightful question. Actually, we have explored this direction to directly orthogonalize the unrolled convolutional operator as in [24]. We find it difficult to do the orthogonalization because the unrolled convolutional operator is restricted (e.g. most of the elements should be 0; values in some locations should be the same). These restrictions cannot be met in the results of re-parametrization. Thus, we cannot get an orthogonal operator with our re-parametrization technique here.
>
> > [Q10] **Possibility of quantization** (it sounds impossible to quantize your network to INT8, INT4 or even binary as in other CNN compression works)
>
> **Response**: We thank the reviewer for pointing out the possibility of quantization. We believe it is possible to quantize our network for the inference stage, as the instability comes from Newton’s iteration which is only required during training. After we pre-calculate the weights from Newton’s iteration, we can quantize them and perform similarly to other compressed networks.
>
> > [Q11] **Model architecture** (the experiments are only conducted on LipConvnet which is NOT conventionally used in practice. People in the field know very well that robustness is very much architecture-dependent… This in turn renders the LipConvnet characterization of this work purely theoretical yet, frankly, with little practical value as the generalization of the results to the more common modern architectures is doubtful (ConvNeXt, Transformers, etc.).)
>
> **Response**: Thanks for the comment. We acknowledge that the current study of 1-Lipschitz networks does not easily generalize to arbitrary modern architectures. However, we would like to point out that there are two major differences between designing 1-Lipschitz networks and standard networks. First, standard architectures focus a lot on regularizing the network so that it does not overfit (e.g. with dropout), while in 1-Lipschitz networks we do not need strong regularization because the 1-Lipschitz constraint is already strong enough. Second, standard architectures are carefully designed so that the gradient will propagate in a proper manner (e.g. residual connection, BatchNorm), while 1-Lipschitz networks have an intrinsic gradient-norm-preserving property (see BCOP[14]). Thus, it may not be appropriate to directly utilize modern architectures in the design of 1-Lipschitz networks.
>
> In the table below, we conduct additional experiments on a 1-Lipschitz ResNet18 model to bridge this gap following the suggestion, where we (1) replace all the conv layers with 1-Lipschitz convolution, (2) replace all the residual connections with average and (3) enforce the variance parameter of BatchNorm layers to be 1 so that the model is 1-Lipschitz. We can observe that LOT still outperforms SOC on both CIFAR-10 and TinyImageNet, while the performance of the ResNet18 architecture is not as good as that of LipConvNet. Nevertheless, we agree with the reviewer that better architectures can be explored for 1-Lipschitz networks and we view it as an interesting future work.
>
> | CIFAR-10     | Vanilla acc | $\rho$=36/255 | $\rho$=72/255 | $\rho$=108/255 |
> | ------------ | ----------- | ------------- | ------------- | -------------- |
> | ResNet18,SOC | 66.43%      | 43.00%        | 23.00%        | 9.52%          |
> | ResNet18,LOT | **68.85%**  | **45.46%**    | **25.43%**    | **11.35%**     |
>
> | TinyImageNet | Vanilla acc | $\rho$=36/255 | $\rho$=72/255 | $\rho$=108/255 |
> | ------------ | ----------- | ------------- | ------------- | -------------- |
> | ResNet18,SOC | 23.26%      | 11.64%        | 5.21%         | 2.17%          |
> | ResNet18,LOT | **25.09%**  | **12.83%**    | **5.90%**     | **2.62%**      |

---

> > ### Comment · Reviewer_miXu · 2022-08-06
> > **Nice addition of experiments**
> >
> > Thanks for the detailed responses, with strengthened experiments.

---

> ### Author Response · Authors · 2022-08-02
> **Author Response [2/3]**
>
> > [Q4] **Proof of Orthogonality** (In Sect. 3.2, the semi-orthogonality in the linear operation context is fine, but then its convolution counterpart sounds hand-waving. Is the latter mathematically equivalent to its linearized counterpart? Can you use a toy example to show how to bridge the former to the latter? If they are not directly corresponding, then a proof that $conv(W, W^T)=I_{conv}$ is needed.)
>
> **Response**: We are sorry for the confusion here. In the second paragraph, we are introducing the definition of “orthogonal convolution kernel” as those $W$ such that $W \circ W^T = I_{conv}$. The claim that needs to be proved would be “*if $W$ is an orthogonal conv kernel, then the convolution operation is 1-Lipschitz*”. This can be seen by noticing that the convolution operator is a linear operator and that the linear operator corresponding to the orthogonal kernel $W$ is also orthogonal. Therefore, the convolution with an orthogonal kernel is an orthogonal linear function and hence it is 1-Lipschitz. Further discussion can be found in BCOP ([14]).
>
> > [Q5] **Pre-Calculation of V** (I wonder, once V is trained and fixed, why can’t W be computed exactly in terms of V, and then the convolution be done in one step using W?)
>
> **Response**: We thank the reviewer for pointing out this possibility to improve the inference efficiency. It is possible to compute the exact value of $W$ in terms of $V$. However, such $W$ would be a very large convolution kernel (whose size is the same as the input), which may not be efficient. Therefore, we choose to only pre-calculate the result of Newton’s Iteration and do the FFT during the inference process.
>
> > [Q6] **Theory in Sect. 5** (Sect. 5 on the theory is quite involved in terms of math, while its conclusion/discussion is rather trivial.)
>
> **Response**: Thanks for the comment, and we agree that the conclusion is intuitively true which verifies the correctness of our theorem. In addition, we would like to point out that to prove our theorem/conclusion is non-trivial. In the theory, we show that unlabelled data can indeed help with certified robustness under reasonable assumptions (these assumptions are also adopted in existing works [25]). Such theoretical analysis is one of our main contributions since we are the first to show that unlabelled data helps certified robustness, and this is also verified in our experiments.
>
> > [Q7] **Figure 1** (Why are the features from LOT “more meaningful” than those from SOC? Simply because the patterns have higher amplitudes and fluctuations?)
>
> **Response**: Thanks for the comment. We can observe that the features from LOT contain more patterns and less noise, while those from SOC contain more noise and fewer patterns. We view the former as more meaningful, following the previous works in which the visualization with more pattern and less noise is “more meaningful” (e.g. [1] in our paper). We agree with the reviewer that this claim may be too strong, and we will tone down the argument as “the features from LOT contain less noise and more visualizable patterns” for better qualitative understanding.
>
> > [Q8] **Complex number in FFT** (I suppose the matrices are holding complex numbers? How does that affect the complexity and stability of the Newton iterations? And how do you guarantee real matrices when mapped back to the spatial domain?)
>
> **Response**: Yes, the matrices include complex numbers and we can observe from Fig. 5 in the Appendix that Newton’s iteration is still stable. We thank the reviewer for mentioning the question about whether the result is still real. Indeed, we can prove that the results are still real with the theorem below.
>
> **Theorem** (added to the revised manuscript as Theorem F.1): Say $J \in \mathbb{C}^{m \times m}$ is unitary so that $J^* J = I$, and $V = J\tilde{V} J^*$ for $V \in \mathbb{R}^{m \times m}$ and $\tilde{V}\in \mathbb{C}^{m \times m}$. Let $F(V) = (V V^*)^{-\frac12} V$ be our transformation. Then $F(V) = J F(\tilde{V}) J^*$.
>
> A rigorous proof of the theorem is in Appendix F of the revised manuscript.
>
> **Remark**: We concretize this theorem in practice by letting $J$ and $J^*$ be the Fourier transform and inverse Fourier transform respectively. Then, $F(V)$ is a real matrix by definition, and the theorem tells us that the complex matrix $F(\tilde{V})$ composed with Fourier and inverse Fourier transform equals this real matrix $F(V)$, i.e., we can guarantee it is a real matrix when mapped back to the spatial domain.

---

> ### Author Response · Authors · 2022-08-02
> **Author Response [1/3]**
>
> We thank the reviewer for the insightful comments and suggestions. We have provided our answers below and updated our paper following the suggestions.
>
> > [Q1] **Attacks in other norms** (The derivation of this paper is specific to l2 attacks only, while attacks are conventionally taking $l_1$, $l_2$, and $l_\infty$ norms. In fact, the benchmarked SOC paper also shows empirical robustness in other norms.)
>
> **Response**: We thank the reviewer for pointing to the empirical robustness against other norms. Here, we conduct additional experiments on $\ell_\infty$ PGD attack with $\epsilon=8/255$ against SOC and our approach and we show the robust accuracy as below. We can observe that we still achieve better performance on the LipConvNet compared with SOC. We have updated these results in Appendix E.7 in the revision of our paper.
>
> | \# Layers | 5          | 10         | 15         | 20         | 25         | 30         | 35         | 40         |
> | --------- | ---------- | ---------- | ---------- | ---------- | ---------- | ---------- | ---------- | ---------- |
> | SOC       | 27.18%     | 27.95%     | 27.84%     | 26.17%     | 28.93%     | 27.44%     | 14.09%     | 11.68%     |
> | LOT       | **27.19%** | **28.71%** | **29.10%** | **29.45%** | **29.55%** | **28.92%** | **29.32%** | **28.67%** |
>
> > [Q2] **Performance on Other Datasets** (Only CIFAR10/100 test results are provided, which limits the practicality of the framework…. In fact, do you have larger examples beyond 32^2 input sizes?)
>
> **Response**: Thanks for the comment. In the paper, we provided only CIFAR results for fair comparison purposes since existing 1-Lipschitz networks are all tested on the CIFAR dataset. To further compare LOT with existing works, we conduct additional experiments on LOT and SOC networks on the TinyImageNet dataset (which is $64*64$ and include 200 classes) and provide the results as below. We can observe that LOT still outperforms the existing SOC approach in most cases. Meanwhile, we observe that all 1-Lipschitz models have a performance drop on larger datasets compared with vanilla models, and we leave the breakthrough as future work.
>
> |                   | Vanilla acc | $\rho$=36/255 | $\rho$=72/255 | $\rho$=108/255 |
> | ----------------- | ----------- | ------------- | ------------- | -------------- |
> | LipConvNet-5,SOC  | 30.77%      | 19.74%        | 11.60%        | 6.89%          |
> | LipConvNet-5,LOT  | **32.71%**  | **21.44%**    | **12.96%**    | **7.92%**      |
> |                   |             |               |               |                |
> | LipConvNet-10,SOC | 31.94%      | 21.21%        | 12.80%        | **7.79%**      |
> | LipConvNet-10,LOT | **32.31%**  | **21.22%**    | **12.96%**    | 7.75%          |
> |                   |             |               |               |                |
> | LipConvNet-15,SOC | 32.26%      | 21.36%        | 12.94%        | 7.80%          |
> | LipConvNet-15,LOT | **33.14%**  | **22.21%**    | **13.34%**    | **8.12%**      |
> |                   |             |               |               |                |
> | LipConvNet-20,SOC | 32.44%      | 21.27%        | 12.90%        | 7.63%          |
> | LipConvNet-20,LOT | **33.19%**  | **22.02%**    | **13.42%**    | **8.12%**      |
>
> > [Q3] **Training Instability** (The authors need to deliberately jet up the accuracy to FP64 during training/inference and then trim back to FP32. This is likely due to the inverse operation which can become ill-conditioned. This may also be why only CIFAR10/100 examples are used, without moving on to the larger sizes like 224^2 ImageNet examples?)
>
> **Response**: Thanks for the comment. We agree that the instability is due to the inverse operation, or more precisely, Newton’s Iteration in order to calculate the inverse operation. Therefore, we use FP64 and ensure that the orthogonality is preserved well. However, this is not a problem when scaling to larger input, since the calculation of matrix inverse on the Fourier frequency domain is pixel-wise and only related to the number of channels. As we show above, LOT still outperforms SOC on TinyImageNet which is 64*64.

---

### Author Response · Authors · 2022-08-02
**Revision Summary**

We thank all the reviewers for their comments and valuable feedback. We have made the following major updates following the reviews to further improve our work.

1. We evaluate the 1-Lipschitz models against empirical $\ell_\infty$ PGD attacks and show that LOT still outperforms existing works on the CIFAR-10 dataset in Appendix E.7.
2. We add a theorem and its proof to show that the results of inverse FFT after manipulating the values on the complex domain are still real in Appendix F.
3. We add results on the TinyImageNet dataset (100,000 images with sizes 64*64 and 200 classes) and show that LOT still outperforms existing works in Appendix G.
4. We add results of a 1-Lipschitz ResNet-18 architecture and show that LOT is still the best convolution layer in Appendix H.All updates are highlighted in blue in our revision.

If the manuscript is accepted, all contents in Appendix E.7/F/G/H will be merged into the main text given the extra page limit for the camera-ready version.

---

### Meta-Review · Area_Chair_zXHC · 2022-08-22

**Recommendation:** Accept
**Confidence:** Less certain

**Metareview:**

This paper proposes a training framework named Layer-wise Orthogonal Training (LOT) that aims to train 1-Lipschitz convolution layers effectively and improves the certified robustness of Lipschitz-bounded models. The paper also proves that semi-supervised learning can benefit the robustness of Lipschitz-bounded models.

All reviewers agree that this work is new, and the empirical improvement is significant. I follow the majority to recommend acceptance.

**Award:**

No

---

### Decision · Program_Chairs · 2022-09-14

Accept